# Synergistic Effect of Quercetin on Antibacterial Activity of Florfenicol Against *Aeromonas hydrophila* In Vitro and In Vivo

**DOI:** 10.3390/antibiotics11070929

**Published:** 2022-07-10

**Authors:** Xianliang Zhao, Xiuying Cui, Yunpeng Yang, Lei Zhu, Li Li, Xianghui Kong

**Affiliations:** College of Fisheries, Henan Normal University, Xinxiang 453007, China; 2015051@htu.edu.cn (X.Z.); 2120183006@stu.htu.edu.cn (X.C.); 2020283041@stu.htu.edu.cn (Y.Y.); zhulei@htu.edu.cn (L.Z.); lily-fish@htu.edu.cn (L.L.)

**Keywords:** *aeromonas hydrophila*, florfenicol, quercetin, synergistic effect, antibacterial activity

## Abstract

The overuse or abuse of antimicrobial drugs in aquaculture, aggravates the generation of drug-resistant bacteria, which has caused potential risks to human health and the aquaculture industry. Flavonoid–antibiotic combinations have been shown to suppress the emergence of resistance in bacteria, and sometimes even reverse it. Here, the antibacterial activity of florfenicol in combination with quercetin, a potential drug to reverse multidrug resistance, was tested against *Aeromonas hydrophila* (*A*. *hydrophila*). Of eleven selected antimicrobial agents, quercetin and florfenicol showed the strongest bactericidal effect, and fractional inhibitory concentration (FIC) indices were 0.28, showing a highly synergistic effect. Then, the antibacterial activities of quercetin and florfenicol against *A. hydrophila* were further tested in vitro and in vivo. Bacterial viability of *A. hydrophila* decreased in a florfenicol dose-dependent manner, about 16.3–191.4-fold lower in the presence of 15 μg/mL quercetin and 0.156 to 1.25 μg/mL florfenicol than in the absence of quercetin, respectively. The cell killing was maximum at 45 μg/mL quercetin in the dose range tested plus 0.156 μg/mL florfenicol. The viability decreased over time during the combined treatment with quercetin and florfenicol by 60.5- and 115-fold in 0.156 μg/mL florfenicol and 0.625 μg/mL florfenicol, respectively. Additionally, the synergistic effect was confirmed by the bacterial growth curve. Furthermore, quercetin and florfenicol had an obvious synergistic activity in vivo, reducing the bacterial load in the liver, spleen and kidney tissues of *Cyprinus carpio* up to 610.6-fold compared with the florfenicol group, and improving the survival rate of infected fish from 10% in the control group to 90% in drug combinations group. These findings indicated that quercetin could potentiate the antibacterial activity of florfenicol against *A. hydrophila* infection and may reduce the use of antimicrobial drugs and improve the prevention and control capability of bacterial resistance.

## 1. Introduction

Aquaculture is becoming an increasingly important source of fish protein available for human consumption. However, infectious diseases by bacterial pathogens are always a hazard and may cause profound economic losses and problems for human health [1]. *Aeromonas hydrophila* (*A*. *hydrophila*) is a well-known opportunistic pathogen for fish, reptiles, amphibians and humans, and is widely distributed in freshwater, estuarine, and saltwater environments [2]. This organism is a recognized cause of motile *Aeromonas* septicemia (MAS), in which the typical disease signs were external hemorrhages, inflammation and ulcers in aquatic animals and diarrhea in mammals, and the mortality rates can be very high [3,4,5]. Therefore, there is a need for cost-effective approaches for preventing or managing *A. hydrophila* infections in aquaculture.

One approach has been the use of antimicrobial agents, pesticides, and disinfectants to control pathogenic bacteria in aquaculture practices, which is considered to be the most effective approach and has greatly enhanced the survival of infected fish. However, the misuse and overuse of antimicrobial agents for disease treatment, especially since it has become common to increase the use of prophylactic antimicrobial agents, has caused the emergence of multidrug resistant (MDR) bacteria, resulting in a significant decrease in the therapeutic efficiency of antimicrobial agents. Recently, MDR *A. hydrophila* strains have been reported in many countries. Stratev et al. [6] tested *A. hydrophila* strains isolated from different food samples are resistant to a broad range of antimicrobial agents, including tetracycline, quinolones and β-lactams. De Silva et al. [7] reported that *A. hydrophila* isolated from shellfish was resistant to multiple antimicrobials including gentamicin, polymyxin-B, tetracycline and β-lactams. Moreover, the presence of MDR *A. hydrophila* strains is a potential threat to human health because drug resistance in bacteria may be spread among bacteria and water environments [8,9]. To address the problems of overuse of antimicrobial agents, novel approaches with environment-friendly solutions that prevent or revert bacterial resistance attract great attention.

Natural compounds and certain dietary agents, such as some plant flavonoids show potential antibacterial activity and manifest the ability to enhance the action of multiple antimicrobial agents [10]. Quercetin is a type of flavonoid widely presents in a wide variety of fruits and edible plants and exhibits a variety of biological activities. It is well recognized that quercetin has anti-bacterial, anti-viral, anti-fungal and anti-cancer effects [11,12,13]. Despite multiple biological activities associated with quercetin, the broad-spectrum antibacterial effect has been used to prevent and treat various infectious bacterial diseases since ancient times [14,15,16]. Many studies have shown that quercetin also has strong antioxidant and anti-inflammatory effects, and a great potential to enhance immunity in humans and animals, such as promoting disease resistance in zebrafish and common carp [17,18]. Our previous study has reported that some small compounds could modulate the susceptibility of bacteria and enhance the antibacterial activity of antimicrobial agents [19]. This approach can improve the antibacterial effectiveness of antimicrobial agents and reduce the development of MDR strains in aquaculture. 

In the current study, we examine the antibacterial activity of quercetin with different antimicrobial agents; the synergistic effect of quercetin in combination with florfenicol was investigated via bactericidal efficacy, bacterial growth of *A. hydrophila* and disease resistance in *Cyprinus carpio.* The investigation provides alternative novel strategies to prevent and control MDR bacteria and also helps in controlling the emergence of bacterial resistance through the relational use of antimicrobial agents.

## 2. Results

### 2.1. Quercetin Improved the Bactericidal Efficacy of Different Antimicrobial Agents

In order to examine the antibacterial activity of different antimicrobial agents in the presence of quercetin, the bacterial viability was calculated by co-incubating quercetin and antibiotic with *A. hydrophila*. Six different antibiotic classes including eleven antimicrobial agents were used to test the modulation effect of quercetin: beta-lactam (ampicillin, AMP; ceftazidime, CAZ; cefazolin, CZO and cefoperazone, CFP), aminoglycosides (gentamicin, GEN; neomycin, NEO and netilmicin, NET), polymyxins (polymyxin B, PB), lincosamides (clindamycin, DA), quinolones (enrofloxacin, ENR), and amphemicols (florfenicol, FFC). The results showed that bacterial viability decreased by 2.20–34.80-fold in the presence of 30 μg/mL quercetin and antibiotic, compared to the presence of the antibiotic alone (Figure 1). The highest antimicrobial activity was observed in quercetin and florfenicol combinations. However, 30 μg/mL of quercetin exhibits weak antimicrobial activity against *A. hydrophila* (*p* > 0.05).

### 2.2. Antimicrobial Activities of Quercetin and Florfenicol

The antimicrobial activities of quercetin and florfenicol against *A. hydrophila* were evaluated by minimum inhibitory concentration (MIC) values. From Table 1, the MIC value of florfenicol against *A. hydrophila* was observed to be 2.5 μg/mL, and it was 360 μg/mL for quercetin. When quercetin was combined with florfenicol, the fractional inhibitory concentration (FIC) was 90 + 0.078 μg/mL. In particular, quercetin increased the sensitivity of *A. hydrophila* to florfenicol from 2.5 μg/mL to 0.078 μg/mL. The FIC indices of quercetin and florfenicol against *A. hydrophila* were 0.28; synergy is considered at an FIC index ≤ 0.5. Thus, the quercetin and florfenicol combination showed synergistic activity following the description by Marques [20].

### 2.3. Synergistic Bactericidal Effect of Quercetin and Florfenicol against A. hydrophila In Vitro

In order to explore the bactericidal effect of florfenicol in the presence of quercetin, 30 μg/mL of quercetin was combined with different concentrations of florfenicol co-incubating with *A. hydrophila*. The results showed that bacterial viability decreased in a florfenicol dose-dependent manner: Compared with the control group without quercetin, the synergistic bactericidal effect of the combined drug increased by 16.3-, 37.3-, 77.9- and 191.4-fold with increasing doses of florfenicol from 0.156 to 1.25 μg/mL plus 30 μg/mL quercetin, respectively (*p* < 0.01) (Figure 2A). It was also observed that the synergistic bactericidal effect of 0.156 μg/mL plus 30 μg/mL quercetin was better than that of 1.25 μg/mL florfenicol alone. 

When bacteria were incubated with 15–180 μg/mL of quercetin alone, the survival rate of *A. hydrophila* decreased with an increasing dose of quercetin. However, the survival rate of the combined drug was first decreased and then increased with the increase in quercetin concentration. When the quercetin concentration was as low as 15 μg/mL, the survival rate of *A. hydrophila* decreased by 11.0-fold after co-incubation with 0.156 μg/mL florfenicol compared with 0.156 μg/mL florfenicol alone (*p* < 0.01). Specifically, the bactericidal efficiency was the highest, with 64.8-fold at 0.156 μg/mL florfenicol plus 60 μg/mL quercetin, compared with 0.156 μg/mL florfenicol alone (Figure 2B). 

Furthermore, the synergistic effect was investigated with 30 μg/mL quercetin plus 0.156 μg/mL or 0.625 μg/mL florfenicol for increased exposure time. The results in Figure 2C show a time-dependent bactericidal effect, cell viability decreased over time during the combined treatment with quercetin and florfenicol by 60.5- and 115-fold, respectively, over the course of 8 h. 

### 2.4. Effect of the Combination of Quercetin and Florfenicol on Bacterial Growth

In order to explore the effect of the combination of quercetin and florfenicol on bacterial growth, *A. hydrophila* was treated with 30 μg/mL quercetin, 0.156 μg/mL florfenicol, and 30 μg/mL quercetin plus 0.156 μg/mL florfenicol. Compared with the control, the growth of *A. hydrophila* was not significantly inhibited by 0.156 μg/mL florfenicol, and 30 μg/mL quercetin had a certain inhibitory effect on the growth, especially in the logarithmic growth phase at 2–5 h. However, the growth of *A. hydrophila* decreased significantly with 30 μg/mL quercetin plus 0.156 μg/mL florfenicol compared with florfenicol or quercetin alone (Figure 3). Though there was no significant difference or only a slight decrease in the growth patterns when *A. hydrophila* cells were treated with low doses of florfenicol or quercetin, a combination of quercetin and florfenicol profoundly inhibited *A. hydrophila* growth. These results indicated that quercetin significantly improved the bactericidal activity of florfenicol against *A**. hydrophila* in vitro.

### 2.5. Quercetin Improved the Ability of Florfenicol to Eradicate Bacteria

The in vivo therapeutic effect of quercetin and florfenicol was evaluated in common carp (*Cyprinus carpio* L.) infected with *A. hydrophila*. For fish infected with *A. hydrophila*, 2.5 or 10 mg/kg florfenicol and/or 10 mg/kg quercetin were used to control the bacterial infection. The results showed that the synergy of quercetin and florfenicol significantly reduced the bacterial load in all three tissues (*p* < 0.05). Compared with the untreated group, the bacterial load in the fish liver of 2.5 or 10 mg/kg florfenicol plus 10 mg/kg quercetin was reduced by 5.3- and 16.8-fold, respectively (Figure 4A). In spleen tissue, the bacterial load was significantly reduced by about 13.9- and 105.7-fold in the two combination groups, respectively (Figure 4B). In kidney tissue, the bacterial load was significantly reduced by about 138.0- and 610.6-fold in the two combination groups, respectively (Figure 4C). In particular, we found that the bacterial load in the 10 mg/kg quercetin plus 2.5 mg/kg florfenicol group was lower than that in the 10 mg/kg florfenicol group in all three tissues (*p* < 0.05). However, with 2.5 or 10 mg/kg florfenicol or 10 mg/kg quercetin treatment alone, there were no significant changes in bacterial load, except for the treatment with 10 mg/kg florfenicol in spleen tissue. That is to say that quercetin could strengthen the ability of florfenicol to eradicate bacteria and reduce the use of florfenicol by at least four-fold under the same or an even better therapeutic effect (Figure 4). The above results showed that quercetin and florfenicol have obvious synergistic activity in vivo.

### 2.6. Effect of Quercetin and Florfenicol on the Anti-Infection Ability

On the basis of bacterial eradication in vivo, the anti-infection ability of quercetin and florfenicol in infected fish was assessed based on the survival rate. Fish were injected with quercetin and/or florfenicol at 1 h and 12 h after challenge with 0.1 mL 5.0 × 10^7^ colony-forming unit (CFU)/mL *A. hydrophila.* The results showed that fish began to die after 12 h, and the mortality was consistent, about 90%, after 36 h in the control group. The therapeutic efficacy of quercetin and florfenicol treatment alone was not obvious, the death of experimental fish began to occur at 12 h and the survival rate was 20% and 30%, respectively. For quercetin and florfenicol combination treatment, the survival rate of fish reached about 90%, and the death of experimental fish began to occur at 24 h (Figure 5). Hence, the co-administration resulted in a therapeutic efficacy of up to 88.9%, whereas the therapeutic efficacy of the single drug was only 11.1% and 22.2% for quercetin and florfenicol treatment, respectively. Combined with the results of bacteria eradication in three tissues of *C. carpio*, it was speculated that quercetin was capable of strengthening the ability of florfenicol against *A. hydrophila* infection in vivo. 

## 3. Discussion

The emergence of MDR bacteria in aquaculture decreases the therapeutic efficiency of antimicrobial agents and blocks the healthy development of aquaculture [21]. So, it is believed that novel approaches are essentially required to prevent and manage bacterial resistance in a cost-effective and environment-friendly manner. Natural compounds, such as flavonoids have a broad-spectrum antibacterial effect against pathogens, including *Escherichia coli*, *Enterococcus faecalis*, *Pseudomonas aeruginosa*
*(P**. aeruginosa**)* and *Staphylococcus aureus*
*(S**. aureus**)* [22,23]. In particular, some flavonoids have the potential ability to enhance the action of multiple antimicrobial agents, such as rifampicin and fusidic acid, and thus could be an effective strategy against MDR bacteria [24,25,26]. Quercetin is one of the most widely distributed plant flavonoids that have the advantage of no or low cytotoxicity, highlighting its higher safety index in aquaculture [27]. However, despite the multiple biological activities of quercetin being reported, its antibacterial activity against aquatic pathogens is unknown. So far, little literature is available on demonstrating the synergistic effects of quercetin and florfenicol against *A. hydrophila*. 

In the current study, the antibiotic with quercetin combinations against *A. hydrophila* showed encouraging results. Among the antimicrobial agents tested in combination, the bacterial viability with different antimicrobial agents varied and the quercetin–florfenicol combination showed the highest antimicrobial activity. Florfenicol, a broad spectrum bacteriostatic antibiotic belonging to the amphenicol class, is widely used in farm animals for the treatment of various infections; due to the inappropriate use of florfenicol in aquaculture, antibacterial resistance is becoming increasingly serious [28], which leads to florfenicol treatment being ineffective to control this disease. Our results showed that quercetin increased the sensitivity of *A. hydrophila* to florfenicol from 2.5 μg/mL to 0.078 μg/mL and the FIC value affirmed the synergistic activity. These results showed that the combination therapy of quercetin and florfenicol with a synergistic or additive interaction is a promising therapeutic approach. Numerous studies have evaluated the broad-spectrum antimicrobial activities of quercetin against numerous kinds of infections caused by MDR bacteria, but only a few reports showed the synergistic effects of quercetin and antimicrobial agents against *P. aeruginosa*, *Staphylococcus pyogenes*, and *S. aureus* [29,30,31,32]. The activities were tested at a low dose of the quercetin–florfenicol combination against *A. hydrophila* in vitro and in vivo. This work showed that quercetin significantly enhanced the bactericidal capability of florfenicol and inhibited bacterial growth. Furthermore, the combination of quercetin and florfenicol significantly reduced the bacterial load of *A. hydrophila* in different tissues and improved the disease resistance of fish against *A. hydrophila* infection. Interestingly, the effects of quercetin and florfenicol are perhaps similar to the effects of rutin (Vitamin P) on *A. hydrophila* infection [33]. It is indicated that the addition of quercetin as a drug in combination with florfenicol enabled reducing the excessive use of antimicrobial agents, giving similar or better results compared to the antimicrobial agent only monotherapy, thus blocking the occurrence of bacterial resistance. However, single-animal treatment might not be realistic in aquaculture in this study, feeding fish with a diet containing quercetin and florfenicol might represent a viable solution. In our previous studies, exogenous metabolites and antimicrobial agents were injected or orally administrated, and the therapeutic effects are similar [19,34]. So, it is believed that quercetin was capable of strengthening the ability of florfenicol either orally, or by i.p. injection against *A. hydrophila* infection. 

Quercetin has been reported to act as a cell wall and cell membrane inhibitor [35], the possible mechanism for this improvement of antibiotic activity by quercetin may be due to the fact that the cell wall biosynthesis in bacteria is probably inhibited, which, in turn, increases the sensitivity of *A. hydrophila* to florfenicol. Another possible mechanism can be the suppression of the efflux pump in bacteria. Quercetin could affect the efflux pump via potassium leakage, thus showing a good inhibitory effect against methicillin-resistant *Staphylococcus aureus* (MRSA) [32]. However, further investigations are needed to explore the possible mechanisms for this improvement by quercetin.

In the aquaculture industry, the resistance of various bacteria is increasing. In order to reduce the occurrence and harm of bacterial resistance, accelerating the development of new effective disease prevention and control technologies, and finding new compounds that can enhance the antibacterial activity of old antibacterial drugs on drug-resistant bacteria has become an important research field. These studies mainly corroborated each other through in vivo and in vitro experiments. It was found that the combination of quercetin and florfenicol can significantly reduce the bacterial load and enhance the anti-infection ability of *C. carpio* to *A. hydrophila*. In an environment where the resistance of aquaculture bacteria is becoming more and more serious, it is of great significance to improve the therapeutic effect of antibacterial drugs, reduce the use of antibacterial drugs, reduce pollution to the aquaculture environment, and prevent and control aquaculture diseases and green aquaculture. As a natural flavonoid compound, quercetin may have a certain regulatory effect on the immune system of fish and can reverse the resistance of antimicrobial agents; the addition of pollution-free substances from exogenous sources has an absolute advantage in aquaculture. Studying the mechanism of regulating quercetin also provides a basis for the subsequent use of other flavonoids in combination with antimicrobial agents to regulate the molecular mechanisms of fish immunity, which is an effective way to develop new anti-drug-resistant bacteria.

In conclusion, this is the first report that quercetin improved the antibacterial effectiveness of florfenicol against *A. hydrophila* and showed obvious synergistic activity when used in combination. These results describe the capability of quercetin to enhance antibiotic activity across a wide range of antimicrobial agents, including florfenicol, inviting a future study of its use as an antibiotic adjuvant that can be used in a wider clinical setting. Our findings also validate that quercetin could strengthen the ability of florfenicol at low concentrations to eradicate bacteria and enhance the disease resistance against *A. hydrophila* infection in *C. carpio*, indicating their promising role in preventing disease outbreaks in aquaculture. Therefore, the present study reports a bright prospect for the prevention and treatment of fish diseases, which will be a benefit to green and healthy aquaculture.

## 4. Materials and Methods

### 4.1. Bacterial Strains and Culture Conditions

*A. hydrophila* (named Ah01) used in the experiments was isolated and stored in our laboratory, and the strain showed high resistance to several major antimicrobial agents and strong pathogenicity in *C. carpio* [36]. The bacteria were grown in a brain heart infusion (BHI) plate (Sangon Biotech Co., Ltd., Shanghai, China), and single colonies were picked and inoculated into BHI medium at 30 °C overnight. Then, cells were harvested by centrifugation at 8000 rpm for 5 min, washed and resuspended with 0.85% saline, and adjusted the optical density at *λ =* 600 nm (OD_600_) to 0.2 for bactericidal assays and susceptibility tests.

### 4.2. Quercetin and Antibiotic Bactericidal Assays In Vitro

The antibiotic bactericidal assay was carried out as previously described [37]. Briefly, a single colony of *A. hydrophila* was grown in 100 mL of BHI medium overnight at 30 °C. An appropriate amount of bacterial culture was centrifuged at 8000 rpm for 5 min, and the pellet was washed three times with 0.85% saline and re-suspended in an M9 minimal medium containing 10 mM sodium acetate, 1 mM MgSO_4_ and 0.1 mM CaCl_2_. Subsequently, an appropriate amount of quercetin and antibiotic drugs were added to the bacterial suspension and incubated at 30 °C for 6 h. To determine bacterial loads, a 0.1 mL sample was collected at a specified time point and serially diluted. An aliquot of 10 μL of each dilution was plated on BHI agar plates and incubated at 30 °C overnight. The colonies were counted and CFU/mL was calculated. The number of bacterial colonies in samples that were not treated with antibiotics or quercetin was considered the initial number of bacteria. The survival rate of bacteria (% survival) was defined as the percentage of bacterial CFU of the antibiotic- and/or quercetin-treated samples compared with the initial bacterial CFU.

### 4.3. Determination of FIC Based on MIC Determination

The minimum inhibitory concentration (MIC) was developed to measure bacterial susceptibility to florfenicol in the presence of quercetin. Briefly, bacteria were cultured in BHI medium overnight, cultures were collected by centrifugation at 8000 rpm for 5 min and washed three times with sterile saline. Next, 0.1 mL of the bacteria with 1 × 10^5^ CFU were added to each well of a 96-well microtiter polystyrene tray (Sangon Biotech Co., Ltd., Shanghai, China). Additionally, 0.1 mL of various fractional concentrations of quercetin and florfenicol was added to each well. The mixtures were incubated at 30 °C. MIC was determined for 16 h according to the lowest antibiotic concentration that ensures visible bacteria. To reduce errors caused by quercetin precipitation, MIC results continue to be determined with bacterial plate count. The interaction between quercetin and florfenicol was calculated by the fractional inhibitory concentration (FIC) index of the combination as previously described [20]. In summation, the FIC index (FICI) represents the sum of the FICs of each drug tested, where the FIC is determined for each drug by dividing the MIC of each drug when used in combination with the MIC of each drug when used alone. Based on the Marques et al. theory, an FIC index lower than 0.5 indicates synergy, because less drug would be required in order to produce the same effect as the drugs alone [20].

### 4.4. Bacterial Survival Growth Curve

One milliliter of *A. hydrophila* in the late logarithmic phase was incubated in 100 mL of fresh medium, and the cells were treated with the indicated concentration of quercetin and/or florfenicol. The optical density (OD) values were determined by reading the absorbance at *λ =* 600 nm wavelength at different time points. At least three biologic replicates were performed. 

### 4.5. Bacterial Eradication Assay In Vivo

*C. carpio* L. was obtained from a local breeding corporation in Zhengzhou. Fish with an average weight of 20 ± 5 g and length of 10.5 ± 2.3 cm were reared in a 250 L aerated water tank. Before the experiment, four fish were randomly selected, and the liver, spleen and kidney tissues were selected for routine bacteriological tests to ensure there was no previous infection among the experimental population. These fish were intraperitoneally injected with 0.1 mL of the bacteria with 1 × 10^7^ CFU. After 24 h post-infection, all fish were randomly divided into six groups, each with four tails, namely, the control group, 2.5 mg/kg, 10 mg/kg florfenicol group, 10 mg/kg quercetin, 2.5 mg/kg florfenicol plus 10 mg/kg quercetin group, 10 mg/kg florfenicol plus 10 mg/kg quercetin group. After 24 h treatment, the liver, kidney and spleen tissue of the fish were excised, weighed, and homogenized in an appropriate amount of saline. The homogenates were then serially diluted to determine the bacterial load (CFU/g) by using the plate count method, as described above. 

### 4.6. Bacterial Infection and Drugs Therapy

To investigate the effect of the drug therapy on the fish survival rate, one hundred and twenty *C. carpio* were challenged with 0.1 mL 1 × 10^8^ CFU/mL. Then, fish were randomly divided into four groups, with 30 fish per group, including saline as the control group, 10 mg/kg quercetin group, 2.5 mg/kg florfenicol group, 10 mg/kg quercetin plus 2.5 mg/kg florfenicol group. All fish received a single treatment administered intraperitoneally with 0.1 mL solutions. After drug therapy, the mortality of the infected fish was observed every 12 h for 6 days. The anti-infection ability of the drug was calculated using the equation: drug therapy efficacy = [1 − (mortality rate of drug administration group/mortality rate of the control group)] × 100% [34,38].

### 4.7. Data Analysis

The experiment was performed with at least three biological replicates. All graph data were presented as mean ± standard error of means (SEM) and graphed using GraphPad Prism v7 (GraphPad Software, Inc., San Diego, CA, USA). Statistical analysis was performed with a two-tailed Student’s *t*-test for the comparison between the two groups. Differences were considered to be significant at *p* < 0.05 and extremely significant at *p* < 0.01.

## Figures and Tables

**Figure 1 antibiotics-11-00929-f001:**
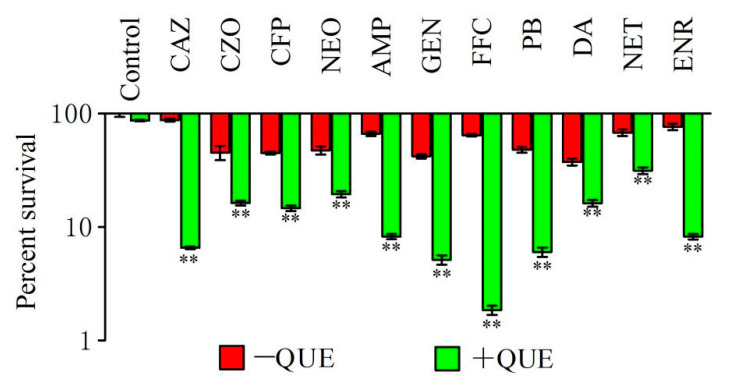
Percent survival of *A. hydrophila* in the presence or absence of quercetin plus the indicated antimicrobial agents. CAZ, ceftazidime, 0.04 μg/mL; CZO, cefazolin, 0.625 μg/mL; CFP, cefoperazone, 0.625 μg/mL; NEO, neomycin, 0.625 μg/mL; AMP, ampicillin, 250 μg/mL; GEN, gentamicin, 0.625 μg/mL; FFC, florfenicol, 0.156 μg/mL; PB, polymyxin B, 25 μg/mL; DA, clindamycin, 25 μg/mL; NET, netilmicin, 0.25 μg/mL and ENR, enrofloxacin, 1 μg/mL; QUE, quercetin. ** *p* < 0.01.

**Figure 2 antibiotics-11-00929-f002:**
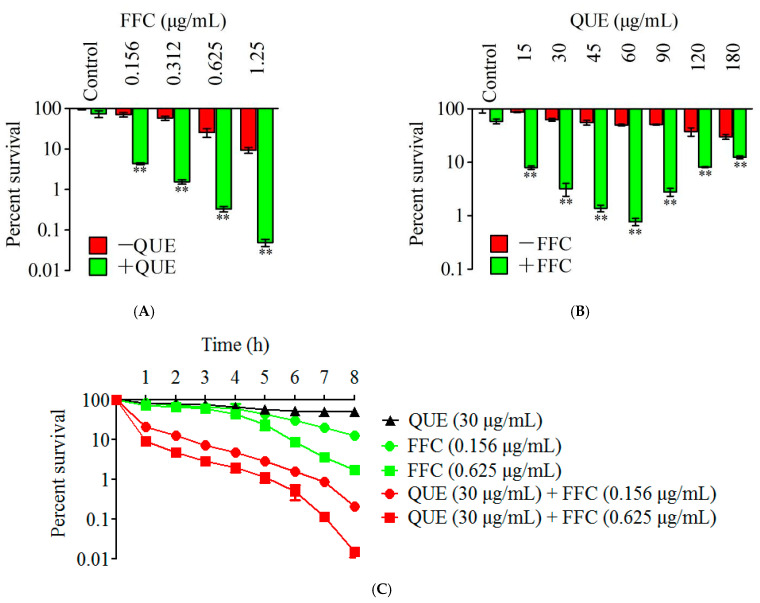
Synergistic bactericidal effect of quercetin and florfenicol against *A. hydrophila* in vitro. (**A**), The bacterial survival rate of 30 μg/mL quercetin combined with different concentrations of florfenicol; (**B**), The bacterial survival rate of 0.156 μg/mL florfenicol combined with quercetin at different concentrations; (**C**), The bacterial survival rate was 0.156 μg/mL or 0.625 μg/mL of florfenicol combined with 30 μg/mL quercetin. QUE, quercetin; FFC, florfenicol. ** *p* < 0.01.

**Figure 3 antibiotics-11-00929-f003:**
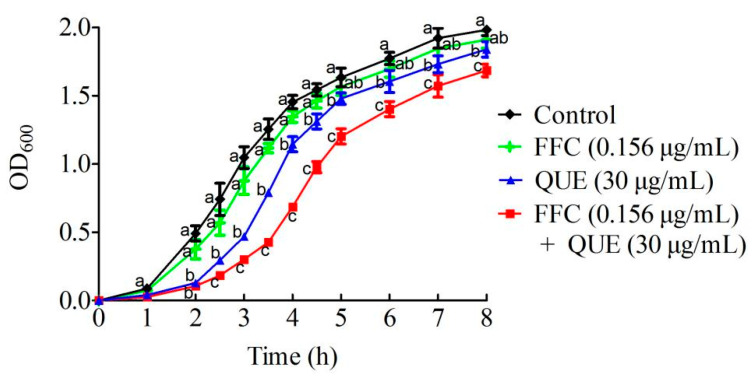
The growth curve of *A. hydrophila* treated with quercetin and florfenicol either alone or in combination. Untreated cells were used as control. QUE, quercetin; FFC, florfenicol.

**Figure 4 antibiotics-11-00929-f004:**
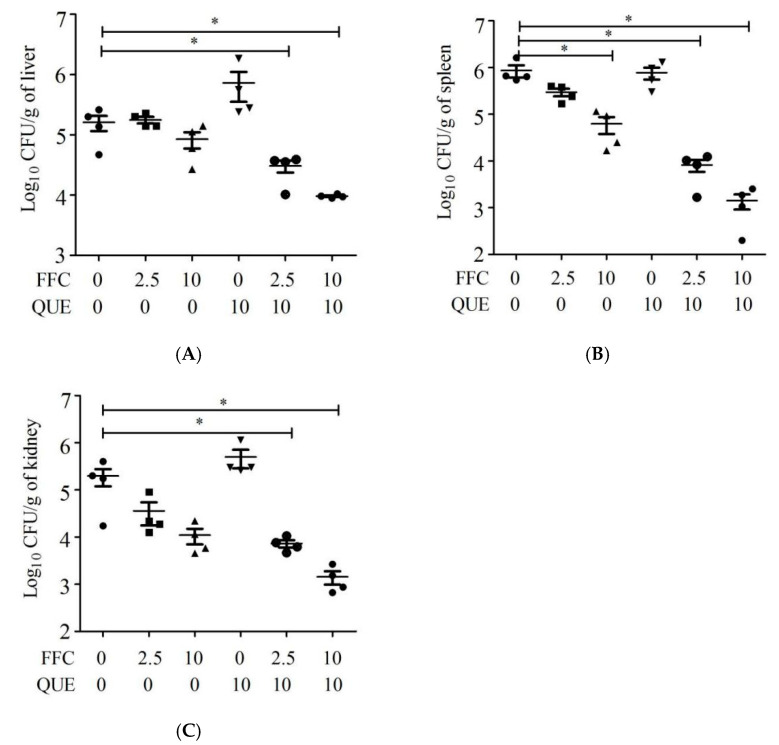
The bacterial load of *A. hydrophila* in liver (**A**), spleen (**B**) and kidney (**C**) of *C. carpio* (*n* = 6) after treatment with quercetin, florfenicol and their combination. QUE, quercetin; FFC, florfenicol. Each point represents data from an individual tissue. * *p* < 0.05.

**Figure 5 antibiotics-11-00929-f005:**
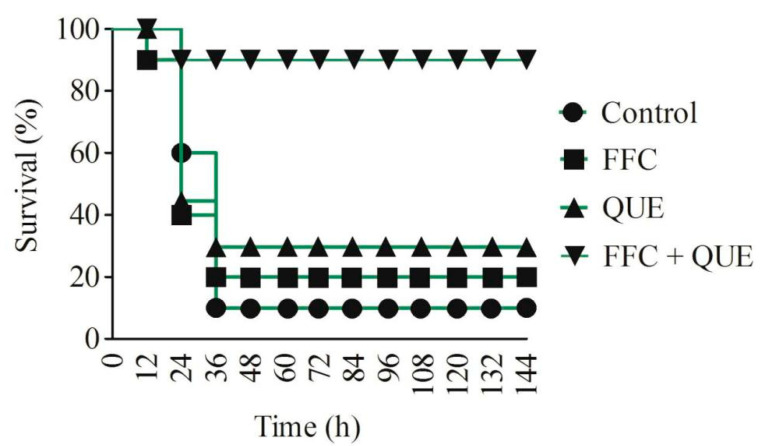
Survival rate of *C. carpio* treated with quercetin, florfenicol and their combination. QUE, quercetin; FFC, florfenicol.

**Table 1 antibiotics-11-00929-t001:** Minimum inhibitory concentration (MIC), fractional inhibitory concentration (FIC) and FIC indexes (FICI) of quercetin and florfenicol against *A. hydrophila*.

Strain	MIC_QUE_ (μg/mL)	MIC_FFC_ (μg/mL)	FIC_QUE+FFC_ (μg/mL)	FICI
*A. hydrophila*	360	2.5	90 + 0.078	0.28

## Data Availability

Raw data are available from authors at request.

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
