# Peer review of "Synergistic Effect of Quercetin on Antibacterial Activity of Florfenicol Against *Aeromonas hydrophila* In Vitro and In Vivo"

_antibiotics, 2022, doi:10.3390/antibiotics11070929_

Round 1

Reviewer 1 Report

"Synergistic effect of quercetin on antibacterial activity of 2 florfenicol against Aeromonas hydrophila in vitro and in vivo"

The paper is a very nice study on synergistic effects of antibiotics to reduce the impact on resistance development and on the environment. I propose to accept the paper with a few minor changes.

l. 48  "However, the misuse or overuse of antimicrobial agents for disease treatment.." Also in particular the prophylactic use causes the development of antibiotic resitance. Please clarify.

l.88 Why is CIP being investigated? Enrofloxacin is approved for the treatment of animals, CIP is only for human use. Please clarify.

ll. 201-214 This paragraph is not really clear to me. The bacteria are not eradicated but reduced in number. Quercetin and florfenicol should not have a protective effect, but serve to fight the infection, i.e. have a curative effect. However, this paragraph repeatedly talks about protective Ability, which cannot be the goal of treatment. Please rephrase.

ll. 239 ff Florfenicol is a bacteriostatic agent that appears to be able to synergistically enhance its effect. This is not discussed here. Please add.

l 263 ff Is single-animal treatment as studied here realistic in aquaculture? Please discuss how the fish should be treated.

Author Response

  1. "However, the misuse or overuse of antimicrobial agents for disease treatment.." Also in particular the prophylactic use causes the development of antibiotic resistance.

Answer: We appreciate this comment. We added a sentence “especially it has become common to increase the use of prophylactic antimicrobial agents” in lines 46-47.  

  1.  Why is CIP being investigated? Enrofloxacin is approved for the treatment of animals, CIP is only for human use. Please clarify.

Answer: Thanks for reviewer’s valuable suggestions. In fact, ciprofloxacin (CIP) and enrofloxacin (ENR) were representative antimicrobial agents of quinolones. Considering that the present study is mainly used for the prevention and treatment of fish diseases, enrofloxacin is more suitable. So, we replace the antibacterial activity of CIP with enrofloxacin (ENR) in the presence of quercetin (Figure 1, lines 85 and 96) in the revised manuscript.

  1.  201-214 This paragraph is not really clear to me. The bacteria are not eradicated but reduced in number. Quercetin and florfenicol should not have a protective effect, but serve to fight the infection, i.e. have a curative effect. However, this paragraph repeatedly talks about protective Ability, which cannot be the goal of treatment. Please rephrase.

Answer: We are very sorry for not describing clearly. We have re-written the description “protective effect” as “anti-infection ability” or “therapy efficacy” in lines 193, 197, 201, 202. In addition, we have rephrased the sentences of 4.6 section in lines 352-356.

  1.  239 ff Florfenicol is a bacteriostatic agent that appears to be able to synergistically enhance its effect. This is not discussed here. Please add.

Answer: We appreciate this comment. We have added the description in lines 227-231 and 233-234.

  1.  263 ff Is single-animal treatment as studied here realistic in aquaculture? Please discuss how the fish should be treated.

Answer: We appreciate this comment. In this study, all fish received a single treatment administered intraperitoneally with 0.1 mL solutions. In fact, single-animal treatment might not be realistic in aquaculture, feeding fish with a diet containing quercetin and florfenicol might represent a viable solution. On our previous studies, exogenous metabolites and antimicrobial agents were infection or oral administration, and the therapy effects are similar (Zhao et al., Aquac Res. 2022; Zhao et al., FEMS Microbiol Lett. 2018; Zhao et al., Fish Shellfish Immunol. 2015). So, it is believed that quercetin was capable of strengthening the ability of florfenicol either orally or by i.p. injection against A. hydrophila infection. We have added the detailed description to better understand the results in Discussion section (lines 247-252).

Reviewer 2 Report

The Authors submitted a paper regarding the synergistic effect of quercetin-florfenicol against Aeromonas hydrophila. The work is interesting and well written. However, the Authors in the experiments tested only A. hydrophila isolates from the Laboratory. For the work to be complete and have scientific validity, the Authors should test also a standard strain. Please, find attached the pdf file with my comments.

Author Response

The Authors submitted a paper regarding the synergistic effect of quercetin-florfenicol against Aeromonas hydrophila. The work is interesting and well written. However, the Authors in the experiments tested only A. hydrophila isolates from the Laboratory. For the work to be complete and have scientific validity, the Authors should test also a standard strain. Please, find attached the pdf file with my comments.

Answer: We appreciate this comment. At first, we carried out this work based on the standard strain A. hydrophila ATCC7966, and we also found that quercetin and florfenicol has obvious synergistic activity in vitro. In our previous study, we compared the pathogenicity, antimicrobial susceptibility and therapy efficacy of A. hydrophila isolate and the standard strain (Zhao et al., J Vet Med Sci. 2019). We found that the LD50 of the A. hydrophila isolate was 2.5 × 108 CFU/fish. However, there was no death or no sign of disease in fish infected with 2.5 × 108 CFU/fish A. hydrophila ATCC7966. Thus, it is difficult to perform the anti-infection ability of quercetin and florfenicol on infected fish in vivo using a standard strain. This is the reason that we performed the experiments tested only A. hydrophila isolates from the Laboratory.

  1.  Please, add A. hydrophila in brackets.

Answer: We appreciate this comment. It has been revised (lines 15 and 37) in the revised manuscript.

  1.  add space

Answer: We appreciate this comment. It has been revised (lines 20), and we also checked other sections in the revised manuscript.

  1.  Please, add the meaning of all the acronyms.

Answer: We appreciate this comment. We have added the meaning of QUE and FFC in lines 96, 140, 156, 190, 207-208.

  1.  Please, add "minimum inhibitory concentration (MIC)"

Answer: We appreciate this comment. We have added the description in line 100.

  1.  Please, add "colony-forming unit (CFU)"

Answer: We appreciate this comment. We have indicated the meaning of the CFU in lines 195-196.

  1.  add P. aeruginosa, S. aureusin brackets, and add "Streptococcus" in full.

Answer: We appreciate this comment. We have added the description in lines 224 and 223.

  1. Please, add methicillin-resistant Staphylococcus aureus (MRSA in brackets)

Answer: We appreciate this comment. We have added the description in lines 216 and 237.

  1. The Authors should test a standard strain in addition to the isolate.

Answer: We appreciate this comment. At first, we carried out this work based on the standard strain A. hydrophila ATCC7966, and we also found that quercetin and florfenicol has obvious synergistic activity in vitro. In our previous study, we compared the pathogenicity, antimicrobial susceptibility and therapy efficacy of A. hydrophila isolate and the standard strain (Zhao et al., J Vet Med Sci. 2019). We found that the LD50 of the A. hydrophila isolate was 2.5 × 108 CFU/fish. However, there was no death or no sign of disease in fish infected with 2.5 × 108 CFU/fish A. hydrophila ATCC7966. Thus, it is difficult to perform the anti-infection ability of quercetin and florfenicol on infected fish in vivo using a standard strain. This is the reason that we performed the experiments tested only A. hydrophila isolates from the Laboratory.

  1. Add manufacturing company

Answer: We appreciate this comment. We have added the description in lines 294 and 319.

  1. add space

Answer: We appreciate this comment. It has been revised (lines 20), and we also checked other sections in the revised manuscript (lines 295, 301, 305 and 308 ).

  1. subscript

Answer: We appreciate this comment. It has been revised (lines 304), and we also checked other sections in the revised manuscript (lines 297).

  1. check text style
    Answer:We appreciate this comment. It has been revised in the revised manuscript.

  1. Line 342, “were” instead of "was".

Answer: We appreciate this comment. It has been revised (line 339) in the revised manuscript.

Reviewer 3 Report

The manuscript deals with analysis of antibacterial activity of florfenicol against Aeromonas hydrophila. The results confirmed the effects in vivo and in vitro. The study is especially relevant for the aquaculture. 

Minor considerations:

- I believe it would be good to define the fractional inhibitory concentration (L106). 

- Figure 2: The meaning of abbreviations.

- Please, add citation for "protective ability of drug" (L363). 

Author Response

The manuscript deals with analysis of antibacterial activity of florfenicol against Aeromonas hydrophila. The results confirmed the effects in vivo and in vitro. The study is especially relevant for the aquaculture.

Minor considerations:

  1. I believe it would be good to define the fractional inhibitory concentration (L106).

Answer: We appreciate this comment. We have added the description of fractional inhibitory concentration in lines 325-330.

  1. Figure 2: The meaning of abbreviations.

Answer: We appreciate this comment. We have added the meaning all the acronyms, including QUE, FFC, MIC and CFU in lines 96, 100, 140, 156, 190, 195-196 and 207-208.

  1. Please, add citation for "protective ability of drug" (L363).

Answer: We appreciate this comment. On our previous studies, we calculated the drug protective rate using this equation (Zhao et al., Aquac. Res. 2022; Zhao et al., J. Fish Dis. 2018). Additionally, according to the comments of other reviewers, we changed “protective ability of drug” to “drug therapy efficacy” in lines 358-360.

Round 2

Reviewer 2 Report

The Authors have satisfactorily addressed my suggestions. I have no further comments.